# Profiling disease burden and *Borrelia* seroprevalence in Canadians with complex and chronic illness

Victoria P. Sanderson[1], Jennifer C. Miller[2], Vladimir V. Bamm[1], Manali Tilak[1], Vett K. Lloyd[3], Gurpreet Singh-Ranger[4], Melanie K. B. Wills[1] *

1 G. Magnotta Lyme Disease Research Lab, Department of Molecular and Cellular Biology, University of Guelph, Guelph, Ontario, Canada, 2 Galaxy Diagnostics, Research Triangle Park, North Carolina, Raleigh, United States of America, 3 Department of Biology, Mount Allison University, Sackville, New Brunswick, Canada, 4 Upper River Valley Hospital, Horizon Health Network, Waterville, New Brunswick, Canada

* mwills@uoguelph.ca

**Data Availability Statement:** All relevant data are within the paper and its Supporting Information files.

## Abstract

Lyme disease, caused by vector-borne *Borrelia* bacteria, can present with diverse multi-system symptoms that resemble other conditions. The objective of this study was to evaluate disease presentations and *Borrelia* seroreactivity in individuals experiencing a spectrum of chronic and complex illnesses. We recruited 157 participants from Eastern Canada who reported one or more diagnoses of Lyme disease, neurological, rheumatic, autoimmune, inflammatory, gastrointestinal, or cardiovascular illnesses, or were asymptomatic and presumed healthy. Intake categories were used to classify participants based on their perceived proximity to Lyme disease, distinguishing between those with a disclosed history of *Borrelia* infection, those with lookalike conditions (e.g. fibromyalgia syndrome), and those with unrelated ailments (e.g. intestinal polyps). Participants completed three questionnaires, the SF-36 v1, SIQR, and HMQ, to capture symptoms and functional burden, and provided blood serum for analysis at an accredited diagnostic lab. Two-tiered IgG and IgM serological assessments (whole cell ELISA and Western blot) were performed in a blinded fashion on all samples. The pattern of symptoms and functional burden were similarly profound in the presumptive Lyme and Lyme-like disease categories. *Borrelia* seroprevalence across the study cohort was 10% for each of IgG and IgM, and occurred within and beyond the Lyme disease intake category. Western blot positivity in the absence of reactive ELISA was also substantial. Fibromyalgia was the most common individual diagnostic tag disclosed by two-tier IgG-positive participants who did not report a history of Lyme disease. Within the IgG seropositive cohort, the presence of antibodies against the 31 kDa Outer Surface Protein A (OspA) was associated with significantly better health outcomes. Previously, this marker has been linked to treatment-refractory Lyme arthritis. Overall, our findings support prior observations of phenotypic overlap between Lyme and other diseases. Seropositivity associated with non-specific symptoms and functional impairment warrants further mechanistic investigation and therapeutic optimization.

**Funding:** M.K.B.W.: G. Magnotta Foundation for Vector-Borne Diseases (www.gmagnottafoundation.com) V.K.L & G.S.R.: The McCain Foundation (https://mccainfoundation.org) G.S.R.: Rotary Woodstock, NB (https://rotarywoodstocknb.com) The funders had no role in study design, data collection and analysis, decision to publish, or preparation of the manuscript.

**Competing interests:** The authors have declared that no competing interests exist.

**Abbreviations:** Bb, Borrelia burgdorferi; CLD, Chronic Lyme disease; ELISA, Enzyme-linked immunosorbent assay; EM, Erythema migrans; FMS, Fibromyalgia syndrome; HMQ, Horowitz Multiple Systemic Infectious Disease Syndrome (MSIDS) Questionnaire; IgG, Immunoglobulin G; IgM, Immunoglobulin M; LD, Lyme disease; MCS$_c$, SF-36 mental component summary, correlated; ME/CFS, Myalgic encephalomyelitis / chronic fatigue syndrome; MS, Multiple sclerosis; PCA, Principal component analy; PCS$_c$, SF-36 physical component summary, correla; PTLDS, Post treatment Lyme disease syndrome; SF-36, RAND 36-Item Short Form Health Survey V 1; SIQR, Revised Symptom Impact Questionnaire (SIQR); TBD, Tick-borne diseases; WB, Western blot.

## Introduction

Although Lyme disease (LD) is a growing public health threat in many parts of the world, there is little consensus on the scope of long-term implications of infection. Caused by spirochetes of the Lyme *Borrelia* complex that are transmitted to a human host through the bite of an infected *Ixodes* s.p.p tick, LD can develop from a local infection into a systemic illness and give rise to diverse and debilitating symptoms. Dermatologic, musculoskeletal, neurological, and cardiac manifestations are among the best documented complications of untreated *Borrelia* infections [1].

The diagnosis of Lyme disease considers an individual's risk factors and exposure potential, clinical presentation, and laboratory findings [2]. An erythema migrans (EM) (target or bull's-eye rash) is the only unique clinical sign early in infection, however the estimated frequency of EM is highly variable (18–80% of cases), and the lesions are heterogenous when present, rendering it an inconsistent indicator of disease [3–6]. Two-tiered serological laboratory testing is currently endorsed in North America and Europe for diagnostic and surveillance purposes. It evaluates the presence of host anti-*Borrelia* IgM or IgG immunoglobulins via a screening ELISA followed by a confirmatory Western blot (WB) when warranted, which is interpreted according to guidelines established by the US Centers for Disease Control and Prevention (CDC). This indirect approach depends on maturation of an adaptive immune response, resulting in low test sensitivity of ~46% over first four weeks of infection [7]. With prompt antibiotic intervention, the majority of patients regain their previous state of health, while 10 to 20% experience lingering or evolving symptoms [8]. Failure to diagnose and treat early in disease has been associated with worse outcomes and therapy resistant manifestations [9]. In the United States, approximately 40% of patients appear to be diagnosed in the late disseminated stage [10]. Recent surveillance efforts suggest that the proportion is similar in Canada, where arthritis, a late manifestation, was identified in 35.7% of reported LD cases [11]. A parallel initiative studying Canadian children found that arthritis accounted for the majority (59%) of LD diagnoses made by a participating paediatrician [12].

If Lyme disease progresses unimpeded or fails to resolve with intervention, it can continue to give rise to a range of multisystem symptoms leading to diagnostic ambiguity and therapeutic uncertainty. Late disseminated Lyme disease, Post-Treatment Lyme Disease Syndrome (PTLDS), and Chronic Lyme Disease (CLD) all describe variations of longstanding, complex, and / or treatment-refractory disease, albeit with distinct clinical implications [2]. The phenotypic heterogeneity of Lyme disease observed in the 1980s prompted its designation as a new "great imitator", after it was found to mimic neurological conditions such as MS and brain tumours [13]. *Borrelia* infection has since been postulated to resemble, initiate, or participate in a variety of complex and chronic health conditions, particularly those of unknown or contested etiology. These include Parkinson's-like movement disorders, dementias and neurodegenerative diseases, as well as autoimmune, inflammatory, and systemic afflictions like fibromyalgia syndrome (FMS) and myalgic encephalomyelitis / chronic fatigue syndrome (ME/CFS) [14–19]. The putative association between these disorders and LD has remained contentious, and indeed the surveillance case definition established by the Centers for Disease Control and Prevention in the 1990s restricted LD to classic objective presentations such as EM or joint effusion. With the addition of a "possible late Lyme disease" case category in 2008 that does not require objective clinical abnormalities, and instead encompasses subjective symptoms such as fatigue and widespread musculoskeletal pain accompanied by reactive IgG serology, syndromic Lyme disease presentations have been described [20]. Although one definition of CLD identifies unresolved *Borrelia* infection as the etiologic agent of longstanding illness [21], the concept has incited controversy, and a dominant clinical narrative maintains that chronic Lyme symptoms are falsely attributed to tick-borne pathogens [22].

Evidently, the role that infection plays in complex and chronic illness is still far from characterized. Pathogenic drivers of disability have recently been underscored by mounting evidence of a causal link between Epstein-Barr virus and multiple sclerosis (MS) [23], as well as the recognition and preliminary characterization of COVID, which refers to the spectrum of sequalae arising from the SARS CoV-2 pandemic virus circulating globally since 2020 [24]. The presentation of long COVID bears many similarities to CLD/PTLDS, ME/CFS, and FMS, highlighting the connection between infectious diseases and protracted syndromic illness [25]. With increasing recognition of this phenomenon and urgent need to ameliorate the impact of chronic diseases, it is of great interest to revisit the potential connection to tick-borne illness. As part of a broader investigation, the current study looked at disease burden and *Borrelia* seroreactivity in individuals in Eastern Canada experiencing a range of complex and chronic conditions.

## Materials and methods

### Participant enrollment & sampling

Research Ethics Board approval was obtained for this cross-sectional study at three collaborating institutions: University of Guelph, Ontario, Canada (REB: 18-07-007), Horizon Health (REB: 2018–2632) and Mount Allison University (REB: 102233), New Brunswick, Canada. The reporting of findings from this observational study conforms to the guidelines and recommendations issued in the STROBE statement. A sample size of 300 individuals was originally planned to ensure diverse and robust representation of people with and without chronic health conditions. However, after enrolling 157 participants, intake was paused in the fall of 2019 for methodological optimization unrelated to the outcomes reported here. We were unable to resume sampling in the spring of 2020 due to the COVID-19 pandemic. All interactions with participants occurred at Upper River Valley Hospital in Waterville, New Brunswick between September 2018 and September 2019. Study enrolment was broadly aimed at English-speaking individuals who identified as healthy or having one or more chronic illness, and was open to anyone who was capable of consenting to research involvement, regardless of the nature of their diagnosis or treatment status. Participants were recruited via invitations distributed to local support and advocacy groups related to complex and chronic diseases (e.g. LD, ME/CFS, FMS, MS) as well as through information posters at the hospital, and via the overseeing physician's practice. Interested individuals were invited to attend a clinic at the hospital, where consent was obtained in writing by the lead physician. After providing a brief medical history to the physician, participants completed three additional questionnaires (described below) to document symptoms, health status, and quality of life. Subsequently, blood was drawn into an uncoated serum vacutainer and shipped to the University of Guelph. Specimens and data were coded at the time of collection, and identifying information was available only to the lead clinician. In eight cases, medical history (including age and sex) were missing; categorial variables (sex, comorbidities) were recorded as unknown, and continuous variables (age, number of diagnoses) were excluded from descriptive statistics. Participants were classified based on the remainder of the available information. Complete questionnaire responses and blood were available from all 157 participants.

### Symptom and quality of life inventories

The Revised Symptom Impact Questionnaire (SIQR) is a generic version of the Revised Fibromyalgia Impact Questionnaire (FIQR), and consists of 21 Likert scale (0–10) questions covering three domains–function, symptoms, and overall impact. A composite score was calculated

for each participant as previously described and validated [26]. The FIQ has previously been used to assess FMS in the Canadian population [27].

The RAND 36-Item Short Form Health Survey (SF-36) V1.0 resolves into eight scales: physical functioning, role limitations due to physical health, role limitations due to emotional problems, energy / fatigue, emotional well-being, social functioning, pain, general health. Normative data for the Canadian population was previously published [28], and served as a comparator for our findings. Individual responses were scored according to the key, and then the components of each aforementioned scale were averaged to yield a value between 0–100, as previously described [29]. Norm-based summary scores for physical (PCS$_c$) and mental health (MCS$_c$) were computed in a three-step process [30]. Scales were first standardized via z-score transformation using Canadian normative data [28], then multiplied by physical and mental health factor scoring coefficients previously generated by an oblique (correlated) factor solution [31]. The products of all eight scales were summed to yield raw aggregate physical and mental health scores, and T-score transformations produced the final PCS$_c$ and MCS$_c$ values [30].

The Horowitz Multiple Systemic Infectious Disease Syndrome (MSIDS) Questionnaire (HMQ) consists of 55 questions across four sections: the Symptom Checklist, Common Lyme Score, Lyme Incidence Scale, and overall health, based around symptom constellations observed in American Lyme disease patients [32]. Scores for three of the sections were directly summed from participant response. However, as per the original version of this questionnaire, the Common Lyme Score was calculated as a binary. Five points were awarded if all of the following symptoms were rated as severe or extremely frequent: fatigue, forgetfulness, joint pain or swelling, tingling, numbness, burning, or stabbing sensations, and sleep disturbances, while no points were awarded for this section if these conditions were not met [32]. According to the proposed interpretation key, scores of 20 and below indicate a low likelihood; 21–45 indicate possibility, and 46+ is a higher probability of tick-borne disease. Symptom cluster scores were developed based on the factor structures (neuropathy, cognitive dysfunction, musculoskeletal pain, fatigue, dysautonomia, cardio respiratory) previously identified [32]. These were calculated using an unweighted loading of the relevant components on the factor [33] to arrive at a value between 0 and 1, or a standardized minimum and maximum score for each factor based on the number of components and range of possible scores for each.

## Serology

Blood was refrigerated after collection at Upper River Valley Hospital and transported on cold packs to the University of Guelph. It arrived within 48 hours of donation and was processed immediately. Liquid content of the red-topped vacutainers was transferred into 15 mL conical tube and centrifuged at 1000 g, 4˚C for 10 min to separate sera and residual cells and debris. Serum was frozen and temporarily stored at -20˚C before being batch shipped to Galaxy Diagnostics (N.C., USA) on dry ice. Two-tiered IgM and IgG *Borrelia* serology was performed on all samples using CLIA-validated, COLA accredited laboratory developed tests, and interpreted according to CDC criteria [34, 35]. Both the first-tier ELISA and second-tier Western blot (WB) assays are run against whole cell derivatives of *Borrelia burgdorferi* (Bb) B31 [36, 37]. Well-characterized serum samples (including paired acute and convalescent serum samples from early Lyme patients with an EM rash, neuroborreliosis, Lyme arthritis or carditis, diseases that mimic LD, and healthy controls from both endemic and non-endemic LD regions) from the CDC's Lyme Serum Repository [38] were included in analytical validation studies. Galaxy's Bb IgM and IgG ELISA and Western Blot test results for these samples compared favorably with results previously obtained by the CDC. Per U.S. CLIA regulations,

Galaxy's IgM and IgG Bb ELISA and WB assays are evaluated through semi-annual proficiency testing (PT) administered through the College of American Pathologists (CAP) PT program (Centers for Medicare and Medicaid Services, 2008). All assay operators must also meet all six CLIA competency components, as assessed on an annual basis (Centers for Medicare and Medicaid Services, 2012). All serological testing was performed in a blinded fashion, and WB outcomes were recorded as individual band reactivities, as well as the overall binary CDC result.

### Data analysis

IBM SPSS Statistics versions 26 through 28 and GraphPad Prism version 9.5.0 were used to calculate summary and inferential statistics. HMQ, SIQR, and SF-36 responses were originally recorded on paper, and then input into Microsoft Excel by the research team. Scores for each of the scales were calculated as described above, and manually entered into SPSS. Serology results were provided digitally by Galaxy Diagnostics.

Normality of data was assessed using the Shapiro-Wilk test, while Levene's test was used to determine homogeneity of variances. The Kruskal-Wallis H test (nonparametric one-way ANOVA on ranks) was used to evaluate differences between continuous variables across more than two groups when the assumptions of classical ANOVA were found to be violated. Our interpretation of Kruskal-Wallis test results was informed by examining the shape of the distribution for each category and variable. Non-uniform distributions do not support the use of medians, and thus the tests presented here are based on mean ranks. In the event of significant findings, Dunn's procedure with Bonferroni adjustment was used for pairwise comparisons.

Owing to small sample sizes in groups of interest, Fisher's exact test was chosen for contingency table analysis. Designs greater than 2x2 were computed as Fisher-Freeman-Halton tests.

The suitability of data for Principal Component Analysis (PCA) was first assessed by Kaiser-Meyer-Olkin (KMO) measure of sampling adequacy and Bartlett's test of homogeneity of variances, performed using the *psych* R package (v2.3.6) in RStudio (v2023.06.1+524) [39]. An overall KMO value of $> 0.6$ and a significant Bartlett's test *p*-value $< 0.05$ were used as criteria to proceed with PCA. PCA was then performed on scaled subsections of the FIQ, SF-36, and HMQ data ($n = 156$) using PCA function from the *FactoMineR* package (v2.8) [40]. Screeplot, a plot of eigenvalues, was used to determine the number of principal components to retain. Components 1 and 2, with eigenvalues $> 1$, satisfied the Kaiser's criterion, and were thus retained for further analysis. The total variance explained by components 1 and 2 was calculated, and the contributions of variables to each component were determined.

Relative sensitivity (positive percent agreement) of individual immunoreactive bands predicting overall IgG two-tier serostatus was calculated as (seropositive individuals with band / total seropositive individuals), while relative specificity (negative percent agreement) is given as (seronegative individuals without band / total seronegative individuals). Overall percent agreement was computed as [(seropositive individuals with band + seronegative individuals without band) / total individuals].

## Results

### Demographics & general trends

Over a one-year pre-pandemic period, 157 participants reporting a spectrum of ailments enrolled in this study. The resulting cohort was 68.2% female, with an average age of 57 years (Table 1). Our recruitment strategy was intentionally broad and permissive to capture authentic, real-world presentations of complex and chronic illness, including intersecting morbidities and risk factors. Most participants (148) declared one or more illness (2.6 conditions on average); only nine were free of any reported conditions.

**Table 1. Clinical findings by intake classification.** Diagnostic category assignment and health metric computation are described in the text. Companion pairwise comparisons of main outcome variables across categories are presented in Fig 1A. Mean ± standard deviation.

| | All n = 157 | Healthy n = 9 | Presumptive LD n = 58 | LD-Like n = 73 | Other Illness n = 17 | p-value[a] |
|---|---|---|---|---|---|---|
| **Age** | 57 ± 14 | 49 ± 18 | 57 ± 14 | 57 ± 13 | 60 ± 16 | 0.327 |
| **Sex** | 68.2% F, 26.8% M, 5.1% unknown | 44.4% F, 22.2% M, 33.3% unknown | 69.0% F, 29.3% M, 1.7% unknown | 76.7% F, 17.8% M, 5.5% unknown | 41.2% F, 58.8% M, 0% unknown | 0.001[b] |
| **Comorbid Dx** | 2.6 ± 1.4 | 0 | 3.1 ± 1.4 | 2.6 ± 1.3 | 1.5 ± 0.8 | <0.001 |
| **SF-36 PCSc** | 29.9 ± 12.23 | 53.90 ± 2.92 | 28.62 ± 11.77 | 27.08 ± 10.16 | 33.68 ± 10.73 | <0.001 |
| **SF-36 MCSc** | 34.6 ± 12.53 | 52.67 ± 5.25 | 32.01 ± 13.28 | 33.14 ± 10.97 | 40.09 ± 9.93 | <0.001 |
| **SIQR** | 44.62 ± 23.39 | 6.76 ± 5.71 | 46.59 ± 24.24 | 49.90 ± 19.70 | 34.73 ± 20.57 | <0.001 |
| **HMQ** | 61.21 ± 31.71 | 13.56 ± 13.09 | 73.31 ± 30.52 | 62.63 ± 28.25 | 39.06 ± 21.64 | <0.001 |
| *Neuropathy* | 0.33 ± 0.24 | 0.04 ± 0.07 | 0.38 ± 0.24 | 0.35 ± 0.23 | 0.20 ± 0.17 | <0.001 |
| *Cognitive* | 0.40 ± 0.26 | 0.05 ± 0.07 | 0.45 ± 0.27 | 0.43 ± 0.23 | 0.27 ± 0.18 | <0.001 |
| *MSK* | 0.59 ± 0.32 | 0.13 ± 0.17 | 0.65 ± 0.31 | 0.65 ± 0.28 | 0.38 ± 0.30 | <0.001 |
| *Fatigue* | 0.56 ± 0.27 | 0.12 ± 0.11 | 0.62 ± 0.27 | 0.59 ± 0.25 | 0.48 ± 0.22 | <0.001 |
| *Dysautonomia* | 0.40 ± 0.26 | 0.11 ± 0.14 | 0.44 ± 0.27 | 0.42 ± 0.25 | 0.31 ± 0.21 | 0.002 |
| *Cardio / Resp* | 0.26 ± 0.21 | 0.06 ± 0.14 | 0.29 ± 0.21 | 0.28 ± 0.21 | 0.16 ± 0.18 | <0.001 |

[a] P-values were generated by Kruskal-Wallis H test comparing mean ranks of presumptive LD, LD-like, other illness and healthy categories, after Shapiro-Wilks test identified non-normally distributed response variables. Arithmetic mean scores are presented here for ease of interpretation; mean ranks and Kruskal-Wallis test statistics are given in S1 Table.

[b] Fisher's exact test was used for discrete data (sex).

Due to our focus on LD and lookalike conditions, we first characterized individuals based on their relative proximity to LD using responses gathered in the enrollment consultation and paperwork to designate categories of "presumptive LD," "LD-like," "other" illness, and "healthy." De-identified participant responses were evaluated by two independent researchers to classify individuals by relevant intake diagnoses. Discrepancies between evaluators when calculating scores or determining assignments were resolved by reviewing the raw data and discussing individual anonymized cases when warranted. Participants were assigned to the "presumptive LD" group if one or more of the following conditions were met: the study physician recorded Lyme disease in the medical history, or the health questionnaires indicated that the individual experienced a tick bite with an erythema migrans rash, and / or received a positive diagnostic test result indicating LD. It should not be assumed that all cases of presumptive LD satisfy consensus criteria of the Centers for Disease Control (CDC) or other guiding institutions, or that the physicians who originally diagnosed the participants used identical criteria. Questionnaire summary scores and laboratory testing conducted as part of the present study were not considered in the intake classification. Additionally, the "presumptive LD" category did not attempt to distinguish past exposure from ongoing infection. The "LD-like" category captured participants with no disclosed history of Lyme disease, who reported diagnoses that share symptoms with late LD, including fibromyalgia syndrome (FMS), myalgic encephalomyelitis / chronic fatigue syndrome (ME/ CFS), neurological disorders (multiple sclerosis (MS), Parkinson's disease) and rheumatic conditions (Sjogren's syndrome, rheumatoid arthritis (RA), systemic lupus erythematosus (SLE), arthralgia). "Other" conditions included cancers, cardiovascular disease (hypertension), as well as respiratory (pulmonary fibrosis, asthma), metabolic (diabetes), gastrointestinal (inflammatory bowel disease (IBD), polyps, constipation, diverticulitis) and psychiatric (depression, anxiety) disorders, also with no record of LD in the medical history. All diagnoses, whether clinical or laboratory, were self-declared through the

medical history collection or the health questionnaires, and not corroborated through medical record review. A participant was considered healthy in the absence of any disclosed conditions at the time of enrollment.

In this classification scheme, individuals were counted only once, and those reporting multiple morbidities were grouped based on the most Lyme-adjacent diagnosis. For example, a participant reporting both LD and FMS would be captured in the presumptive LD category, whereas someone with fibromyalgia and a history of cancer would be considered LD-like on account of the FMS diagnosis.

Overall, 83% of participants were classified as presumptive LD or LD-like (Table 1). There was no significant difference in age between four intake categories (p = 0.327), however the gender balance was skewed in favour of females in the presumptive and Lyme-like categories (p = 0.001 across 4 categories). In all, 65.5% of the presumptive-LD cohort reported a history of tick bite / rash, followed by symptoms, and 48% had previously received a positive diagnostic test result indicating Lyme disease. Only 9% were diagnosed with LD without early objective manifestations or laboratory evidence. The presumptive LD group also reported the highest number of comorbid conditions with an average of 3.1 additional diagnoses (Table 1). Dunn's post-hoc analysis confirmed that participants in the healthy or "other" illness categories had significantly fewer comorbidities than those in the presumptive LD and LD-like categories (p < 0.05).

## Health metrics

At the time of enrollment, participants completed three validated questionnaires that have been used in clinical and research settings to assess the impact of LD and related conditions. Summary scores across all instruments were significantly different between healthy and ill cohorts as revealed by the Kruskal-Wallis H test of mean rank, while more subtle distinctions were observed between the three disease groups (Table 1 and Fig 1). However, LD-like and presumptive LD were not significantly different by any of the questionnaire tools. The Rand 36-Item Short Form Health Survey (SF-36) is a general measure of health-related quality of life, from which we calculated norm-based summary T-scores for physical ($PCS_c$) and mental health ($MCS_c$) (Table 1). Values are relative to a national average of 50 and standard deviation of 10, and higher scores indicate fewer functional restrictions. All participants who were deemed healthy at intake scored at or above the population mean, while 77% of the study population was one or more standard deviations below on $PCS_c$. The mental health composite score ($MCS_c$) was significantly lower on average for Lyme and Lyme-like disease than for other illness (Fig 1A and Table 1).

The FIQR was originally developed to assess burden of illness in fibromyalgia patients [26], although it has since proven useful in post-treatment LD [41], and has been re-released with more generic language as the SIQR. With this instrument, higher scores indicate more severe impairment, which was most evident in LD-Like and presumptive LD cases, but also substantial in other illness (Fig 1A and Table 1). Finally, the HMQ has been used as a screening and assessment tool for LD, yielding a summary score that increases with tick-borne disease (TBD) risk factors, as well as the diversity and severity of symptoms. Cohort scores on this instrument were statistically indistinguishable between the presumptive LD and LD-like categories, but both groups were significantly different from other illness and healthy categories (Fig 1A and Table 1). Using the factor structure previously identified for the HMQ [32], we then explored symptom clusters captured by this instrument, and found that four of the six clusters (neuropathy, cognitive, musculoskeletal, and cardio-respiratory) differed significantly between other illness and presumptive LD / LD-like, but were indistinguishable between the latter two

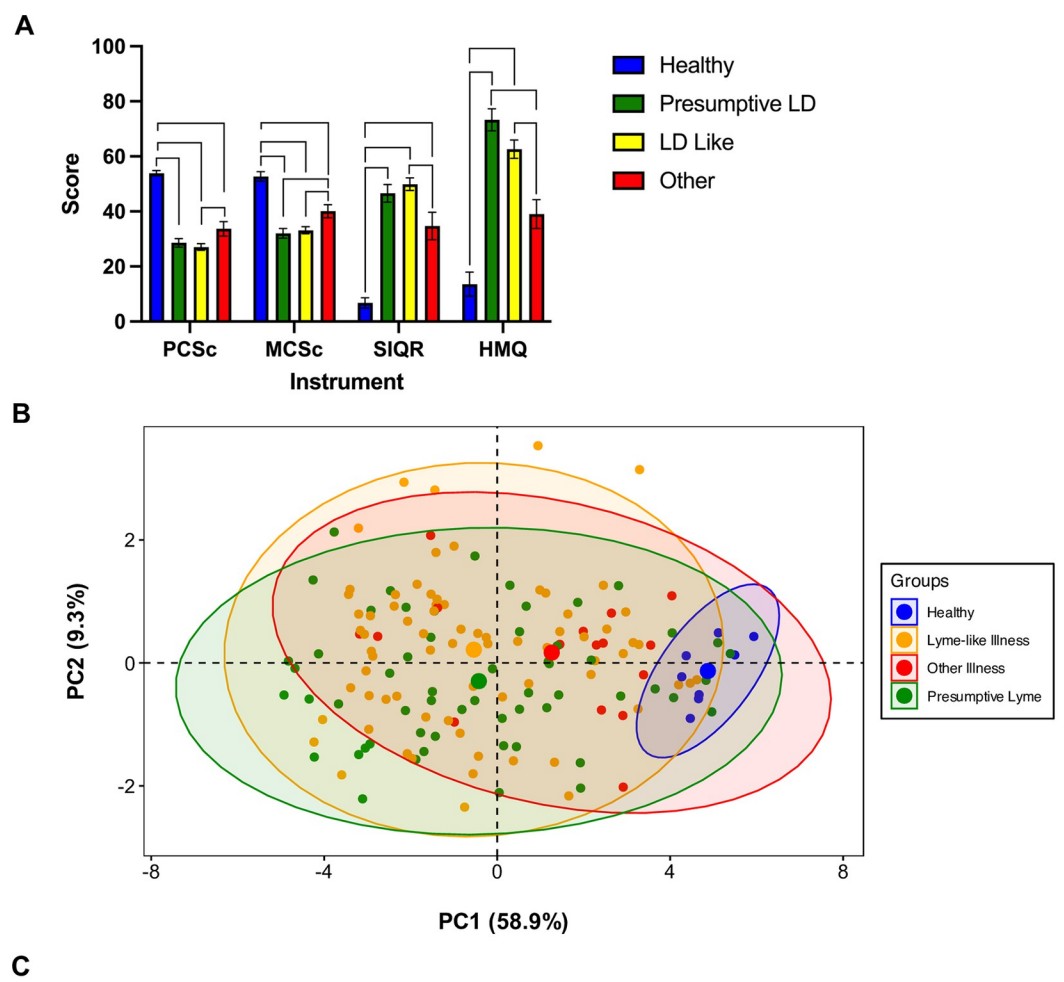

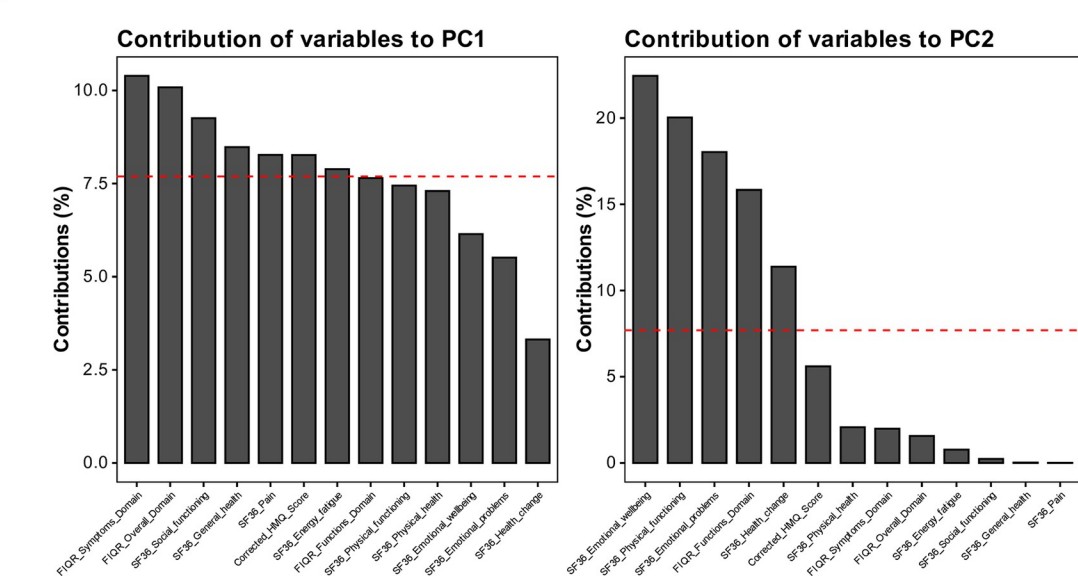

**Fig 1. Exploration of health trends according to participant classification.** (**A**) Average performance of four participant intake categories (colours) on four health indices: SF-36 physical health summary score (PCS$_c$), mental health summary score (MCS$_c$), Revised Symptom Impact Questionnaire (SIQR), and Horowitz Multiple Systemic Infectious Disease Syndrome (MSIDS) Questionnaire (HMQ). For each instrument, mean ranks were compared using the Kruskal-Wallis H test (p-values reported in Table 1; mean ranks and test statistics given in S1 Table) followed by pairwise comparison using Dunn's procedure with

Bonferroni adjustment. Brackets indicate significance at α = 0.05. Mean ± SEM. (**B**) Principal component analysis score plot depicting individual cases (dots) delineated by intake categories (colours). The overall Kaiser-Meyer-Olkin (KMO) measure was 0.91. (**C**) Contributions of each variable to the first two principal components. Bar height corresponds to the amount of variance in the data explained by that variable for the given component. Intake category colours: Blue: healthy; Green: presumptive LD; Orange/yellow: LD-like; Red: other illness.

groups. In clinical settings, HMQ scores have also been stratified in attempt to estimate the likelihood of TBD [32]. 65.6% of study participants assorted into the TBD "probable" category, with questionnaire scores equal to or greater than 45. By intake classification, the HMQ "probable" cohort consisted of 45.6% presumptive LD, 46.6% LD-like, and 6.8% other illness, and 1% healthy. Of those identified as presumptive Lyme upon intake, 81% were designated as "probable" by this instrument, compared to 66% of those in the Lyme-like category.

Individually, questionnaire summary scores did not distinguish well between illness categories. To explore the findings in a more integrative fashion, we performed principal component analysis (PCA) using the subscales of each of the three instruments (Fig 1B). PCA accomplishes unsupervised dimensionality reduction while retaining as much variance in the data as possible, through the strategic generation of composite variables, or components. A two-component solution accounted for 68.2% of the overall variance, and the contributions of the variables to each component are given in Fig 1C. The resulting PCA score plot depicts the individual participants according to health intake category (colour), with ellipses capturing 95% of cases belonging to each respective group (Fig 1B). Although healthy participants cluster in the right quadrant, there remains substantial overlap between disease groups, and no emergent trends.

## Laboratory findings

To evaluate *Borrelia* exposure, we performed IgG and IgM two-tiered serology on the blood of all participants. Conventionally, under CDC guidelines, the second-tier WB is only performed following a positive or equivocal ELISA result, and IgM analysis would only be conducted within 30 days of a tick bite. However, we sought a more comprehensive analysis of the antibody profile for exploratory purposes. Table 2 provides a breakdown of the serology findings by diagnostic intake classification, and indicates that two-tiered positive individuals (henceforth designated seropositive) were identified in all three disease cohorts. Presumptive-LD and LD-like categories, which contained the largest numbers of participants, captured the bulk of the seropositive results. ELISA and WB single-tier positivity also occurred in healthy individuals, and indeed there was no difference between cohorts with respect to the average number of bands on the IgG and IgM WB. Contingency table analysis by Fisher's exact test found no relationship between intake category and serostatus (Table 2). In all, 32% of the study population was negative across both tiers of IgG and IgM analysis.

A bubble plot (Fig 2A) depicts intake category and serostatus of individual participants arrayed by HMQ (x-axis) and SF-36 $PCS_c$ (Y-axis). Seropositive individuals span the range of scores, from lowest (upper left quadrant) to highest (lower right quadrant) disease burden. However, there was just one seropositive participant in the upper left quadrant and only a total of two participants in the section of the plot characterized by < 21 HMQ score. Notably, the few seropositive individuals who declared negligible functional impairment had been classified as presumptive LD upon enrolment, indicating a known history of LD and / or features strongly suggestive of the disease, such as an EM rash. Indeed, most seropositive individuals fell into the section of the plot with an HMQ score higher than 45. At the same time, eight out of nine healthy seronegative participants were clustered in the upper left quadrant and only

**Table 2. Laboratory findings by intake classification.** Within IgG and IgM comparisons, rows and columns are mutually exclusive with the exception of the all participants category. TT: two-tiered; +ve: positive; -ve: negative; equiv: equivocal.

| | | All n = 157 | Presumptive LD n = 58 | LD-Like n = 73 | Other Illness n = 17 | Healthy n = 9 | p-value[a] |
|---|---|---|---|---|---|---|---|
| **IgG** | **TT+ve** | 17 (10.8%) | 8 (13.8%) | 7 (9.6%) | 2 (11.8%) | 0 (0%) | 0.543 |
| | **WB +ve** | 12 (7.6%) | 4 (6.9%) | 4 (5.5%) | 2 (11.8%) | 2 (22.2%) | |
| | **ELISA +ve or equiv** | 50 (31.8%) | 17 (29.3%) | 22 (30.1%) | 7 (41.2%) | 4 (44.4%) | |
| | **-ve** | 78 (49.7%) | 29 (50.0%) | 40 (54.8%) | 6 (35.3%) | 3 (33.3%) | |
| | **Avg # bands (mode)** | 3 (4) | 3 (3) | 3 (4) | 3 (3) | 4 (4) | |
| **IgM** | **TT+ve** | 14 (8.9%) | 7 (12.1%) | 6 (8.2%) | 1 (5.9%) | 0% | 0.372 |
| | **WB +ve** | 20 (12.7%) | 10 (17.2%) | 6 (8.2%) | 3 (17.6%) | 1 (11.1%) | |
| | **ELISA +ve or equiv** | 34 (21.7%) | 10 (17.2%) | 21 (28.8%) | 3 (17.6%) | 0 (0.0%) | |
| | **-ve** | 89 (56.7%) | 31 (53.4%) | 40 (54.8%) | 10 (58.8%) | 8 (88.9%) | |
| | **Avg # bands (mode)** | 1 (1) | 1 (1) | 1 (1) | 1 (1) | 1 (1) | |

[a]Fisher-Freeman-Halton exact test used for discrete categorical data, and compares presumptive LD, LD-like, other illness and healthy

one participant fell into the upper right quadrant (low disease burden but high likelihood of LD based on the HMQ score) (Fig 2A).

S2 Table provides health metrics delineated by serological cohort. On average, IgG seropositive individuals were younger than seronegative participants, and had lower SF-36 physical component summary scores than either the seronegative or IgM positive participants. The musculoskeletal factor of the HMQ was also significantly different between IgG and IgM groups, and between IgM and double-positives (S2 Table). Revisiting the principal component analysis performed above, we re-classified individuals based on serostatus instead of intake category, to assess whether any trends were evident at this level of resolution. As shown in S1 Fig, there is considerable overlap between serostatus groups and no observable patterns, similar to our observation of intake categories.

Overall, 52.9% of IgG seropositive findings were detected outside the presumptive LD intake category, in the LD-like or other illness strata. To determine whether this seropositive cohort with no known history of LD was clinically distinct from the seropositive presumptive LD group, we compared their mean health index scores. As shown in Table 3A, there were no significant differences in SF-36, SIQR, HMQ, or individual factors therein. The only distinguishing feature we noted was a history of tick bite and rash reported by six out of eight participants in the seropositive presumptive Lyme category, of whom five of the six also disclosed a Lyme disease diagnosis.

Considering the number of companion diagnoses disclosed by participants, it was of additional interest to explore conditions co-occurring with positive IgG serology. Within and beyond the presumptive-LD category, fibromyalgia was the most frequent individual condition reported (Fig 2B). Autoimmune and gastrointestinal diagnoses, which both encompassed a range of ailments, were also prevalent.

## Test agreement

Having established basic clinical profiles of the serostatus cohorts, we next sought to understand the alignment between diagnostic tests, and individual tiers of a test. The overall two-tier positivity rate across all participants was ~10% for each immunoglobulin. However, IgG and IgM responders were largely distinct, with little overlap between cohorts (S3A Table). This supports our observation that some clinical characteristics differed between serostatus groups (S1 Table).

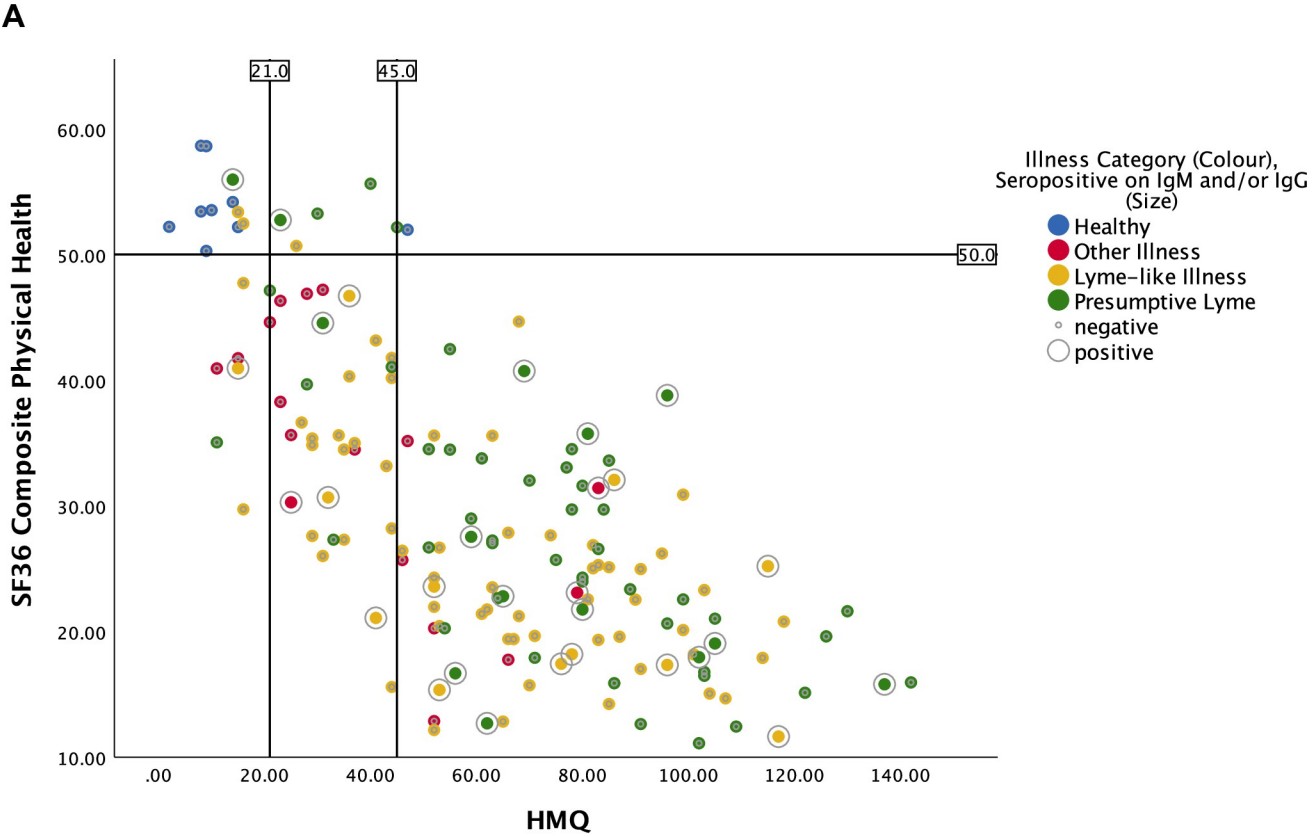

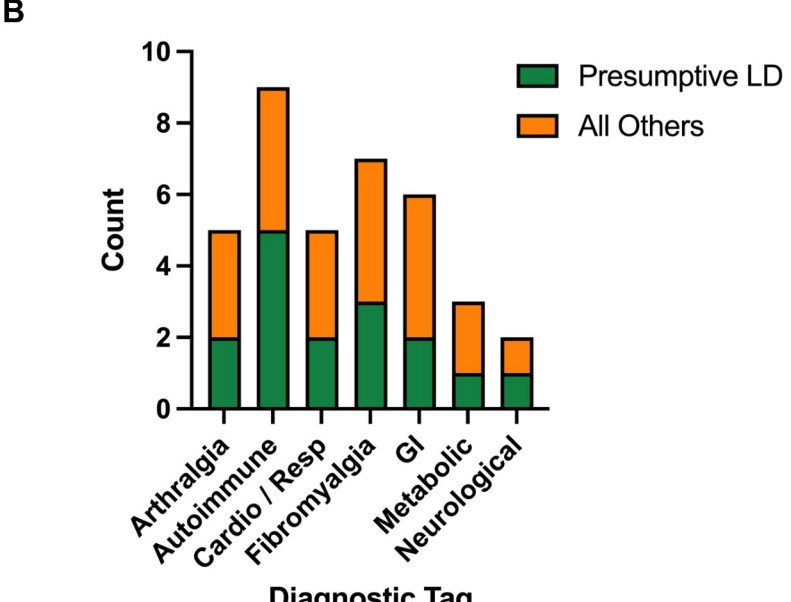

**Fig 2. Illness trends associated with seroreactivity.** (**A**) Distribution of participants according to four parameters: Horowitz Multiple Systemic Infectious Disease Syndrome (MSIDS) Questionnaire (HMQ) score (x-axis), physical burden of disease as captured by the SF-36 physical health summary score (PCS$_c$) (y-axis), point of entry diagnostic category (dot colour), two-tiered serological status, IgG and IgM (dot size). Vertical lines indicate three HMQ classifications based on likelihood of tickborne disease: unlikely (<21), possible (21–45), probable (>45). The horizontal line indicates the mean of SF-36 physical component score in the general population. More profound impairment is indicated by low scores on the PSC$_c$ and higher scores on HMQ. Blue: healthy; Green: presumptive LD; Orange/yellow: LD-like; Red: other illness (**B**) Diagnostic tags disclosed by IgG seropositive participants within and beyond the presumptive LD category. Conditions include rheumatoid and osteo arthritis, Parkinson's disease, Graves' disease, Raynaud's phenomenon,

transverse myelitis, degenerative disk disease, type 1 and 2 diabetes, hives, hypertension, heart problems, fibromyalgia, sleep apnea, William syndrome, Crohn's, ulcerative colitis, constipation, and gastroesophageal reflux. Not captured in the graph are history of cancer and mental illness, each reported by one individual.

As investigated by crosstabulation analysis (S3B and S3C Table) the agreement between test tiers was not significant for IgG or IgM ELISA versus WB. The two-tiered algorithm uses a higher throughput, more sensitive primary assay (ELISA) that must yield a positive or equivocal result to warrant a WB, which refines the specificity [42]. Some degree of attrition between the first and second tier is therefore anticipated. In this study cohort, only 25% of positive or equivocal IgG ELISA and 29% of IgM ELISA subsequently registered a positive WB. On average, participants with negative IgG ELISA results generated three bands on the WB, while ELISA positive and equivocal individuals produced four.

Furthermore, non-reflexive use of the WB allowed us to investigate the prevalence of second tier positivity in the absence of the first. Strikingly, ~ 41% of positive IgG WBs, and 60% of positive IgM WBs occurred in ELISA-negative individuals (S3B and S3C Table).

Relying on participant awareness and recall of a tick bite (with or without signs and symptoms of Lyme disease) was also not a reliable predictor of serostatus (S3D Table). Among participants who reported a previous positive diagnostic test result for LD (n = 28), 32.1% were two-tier seropositive according to IgG and / or IgM. Similarly, 25% of those who reported receiving prior treatment for LD (n = 16) were seropositive in this study.

## Western blot band analysis

Seroreactive bands on a diagnostic Western blot can provide clues about the infecting *Borrelia* strain, host immune response, and disease presentation. To determine whether there was a relationship between the number of immunoreactive bands and illness severity, we performed a correlation analysis using the SF-36 PCS$_c$ score as a readout of health status. No significant association was identified between the number of bands and physical impact of illness (r = 0.040; p = 0.617).

**Table 3. Exploring health status of the IgG seropositive group subdivided by different characteristics.** Burden of illness is compared between seropositive cohorts (**A**) within and beyond the presumptive LD intake category, and (**B**) in the presence and absence of OspA (31kDa) IgG seroreactivity on the Western blot.

| | A) IgG +ve // Intake Class | | | B) IgG +ve // OspA Status | | |
|---|---|---|---|---|---|---|
| | Presumptive LD (N = 8) Mean ± SD | All other (N = 9)[a] Mean ± SD | P-value[b] | 31 kDa OspA Absent (N = 9) Mean ± SD | 31 kDa OspA Present (N = 8) Mean ± SD | P-value[b] |
| **Age (Years)** | 46.13 ± 13.09 | 50.67 ± 14.34 | 0.51 | 48 ± 14 | 49 ± 14 | 0.885 |
| **SF-36 PCSc** | 25.20 ± 15.13 | 22.31 ± 7.53 | 0.62 | 18.57 ± 5.90 | 29.41 ± 13.71 | 0.047 |
| **SF-36 MCSc** | 28.11 ± 15.02 | 30.75 ± 12.15 | 0.70 | 23.03 ± 12.73 | 36.80 ± 9.94 | 0.026 |
| **SIQR** | 58.13 ± 24.59 | 48.57 ± 16.62 | 0.36 | 62.37 ± 16.70 | 42.60 ± 20.51 | 0.045 |
| **HMQ** | 76.25 ± 37.51 | 73.89 ± 27.14 | 0.88 | 95.44 ± 22.84 | 52.00 ± 22.67 | 0.001 |
| *Neuropathy* | 0.34 ± 0.31 | 0.41 ± 0.21 | 0.58 | 0.54 ± 0.22 | 0.20 ± 0.16 | 0.003 |
| *Cognitive* | 0.55 ± 0.35 | 0.46 ± 0.25 | 0.58 | 0.65 ± 0.20 | 0.34 ± 0.31 | 0.026 |
| *MSK* | 0.72 ± 0.41 | 0.74 ± 0.16 | 0.89[c] | 0.86 ± 0.14 | 0.58 ± 0.36 | 0.047 |
| *Fatigue* | 0.56 ± 0.35 | 0.75 ± 0.20 | 0.18 | 0.80 ± 0.25 | 0.51 ± 0.25 | 0.031 |
| *Dysautonomia* | 0.37 ± 0.25 | 0.51 ± 0.20 | 0.21 | 0.56 ± 0.21 | 0.31 ± 0.18 | 0.019 |
| *Cardio / resp* | 0.31 ± 0.27 | 0.41 ± 0.29 | 0.46 | 0.54 ± 0.25 | 0.16 ± 0.14 | 0.002 |

[a] "All other" group contained both LD-like (seven participants) and other illness (two participants) categories

[b] Independent samples 2-tailed t-test with equal variances assumed

[c] Independent samples 2-tailed t-test with equal variances not assumed.

Next, we investigated the potential of each band in the IgG WB to predict the overall serostatus. Relative sensitivity and specificity of each immunoreactive IgG band (positive and negative percent agreement, respectively) are shown in S4 Table. The agreement between band reactivity and overall test result is expressed by the Kappa statistic. Four of the scored IgG bands (23 kDa, 28 kDa, 58 kDa, 66 kDa) as well as OspA (31 kDa), had a significant (non-zero) Kappa. However, the values were still modest, indicating a low agreement between individual band reactivity and overall serostatus. More promiscuous bands (30 kDa, 41 kDa), are also evident in S4 Table.

Although OspA (31 kDa) is excluded from the conventional Western blot interpretation, it was reported here for exploratory purposes. It is one of few borrelial epitopes that has been associated with prolonged complications of LD [43]. We therefore analyzed the health status of the IgG seropositive cohort subdivided by presence or absence of OspA antibodies. Unexpectedly, the anti-OspA-IgG positive group on average reported significantly more favourable health outcomes on all scales (Table 3B).

## Discussion

This study endeavoured to sample a heterogenous real-world population of individuals in Eastern Canada living with complex and chronic diseases, in order to profile the functional burden of illness and evaluate exposure to *Borrelia*. This recruitment strategy has revealed a spectrum of overlapping ailments associated with multisystem dysfunction. The characteristics of the presumptive Lyme disease cohort in this study mirror trends observed internationally with chronic Lyme disease, including profound clinical burden and functional deficits captured by instruments like the SF-36 [14, 21, 44–46]. Women are overrepresented in this group, and many report co-morbidities and / or co-infections. Among our participants, neuropathy, cognitive, musculoskeletal, and cardio-respiratory domains specifically distinguished the Lyme and Lyme-like disease from other illnesses, which is consistent with previous observations that patients rank these symptom clusters as severe or very severe in CLD [46]. While fatigue was also profound, it was prevalent in the "other" disease category as well. At this level of resolution, there were no current clinical features that distinguished the presumptive Lyme and Lyme-like cohorts, highlighting challenges faced by individuals and their healthcare providers in obtaining diagnoses.

Evaluating *Borrelia* exposure via two-tiered serology found ~10% of participants were seropositive by each of IgG and IgM, although the groups were largely distinct. While CDC guidelines restrict the interpretation of *Borrelia* IgM immunoreactivity to the first 30 days of infection, the significance of an IgM response beyond acute *Borrelia* infection is a subject of ongoing investigation. Recently, an association was found between persistently elevated anti-*Borrelia* IgM ELISA values and failure to resolve nonspecific symptoms [47]. Single-tier positives were additionally substantial in our study; 39.5% and 34.4% of participants returned positive or equivocal findings on ELISA or WB for IgG and IgM, respectively. Although it was beyond the scope of the present study, it is of great interest to further investigate specimens for corroborating evidence of *Borrelia* infection. One limitation of serology is that serum from individuals with Epstein-Barr viral infections (mononucleosis), rheumatoid arthritis, *Bartonella* infections, or other spirochetal infections (e.g. periodontitis, syphilis) may exhibit cross-reactivity with *B. burgdorferi* IgM or IgG ELISA and Western blot assays [48–54]. It should also be noted that our serological assays were developed around *B. burgdorferi* sensu stricto strain B31, and their performance with other genospecies has not been evaluated. Related *Borrelia* of interest include *B. afzelii* and *B. girinii*, common in Europe and Asia, and the relapsing fever spirochete *B. miyamotoi*, which is an emerging pathogen in the study area [55]. Acute

and persistent symptoms of *B. miyamotoi* can mimic Lyme disease and other tick-borne illnesses [56], making the case for more comprehensive workups of afflicted individuals.

Indeed, the geography of Lyme and related diseases is shifting, and environmental risk in our recruitment region of the Canadian Maritimes is increasing [57]. In 2008, canine *Borrelia* seropositivity in New Brunswick was less than 1% among sampled animals [58]; five years later, a study from a different group found 6% seroprevalence [59]. According to national surveillance initiatives conducted in 2019, the year of our study enrolment, *I. scapularis* tick positivity in the province was ~13% in specimens submitted by residents, and 21% in those at active sampling sites [60]. While the threat posed by Lyme disease is apparent, there is little published data on human seroprevalence in the area. In the adjacent southeasterly province of Nova Scotia, which boasts the highest documented LD incidence per capita in Canada (Public Health Agency of Canada, 2022) [61], human seroprevalence was estimated at 0.14% based on a random sample in 2012 of 1,855 citizens presumed to be healthy [62]. The same study identified a first tier positivity of 11.6% in the cohort of interest. Meanwhile, an investigation of tick bite recipients to the west in the neighbouring American state of Maine between 2009 and 2013 found that 13.9% were two-tier IgG seroreactive to *Borrelia* [63]. A broader retrospective analysis of specimens collected over two decades from healthy individuals in New England identified *Borrelia* seroreactivity in 9.4% of individuals [64]. Our current study did not adequately sample the general population of New Brunswick to generate an estimate of baseline seroprevalence, and our small control contingent was seronegative. However, six healthy individuals in this group with no reported history of LD returned single-tier IgG positive results (ELISA or WB). Considering the existing threat in New Brunswick and the proximity of the province to regions of the Maritimes / Eastern Seaboard that are deemed higher risk for Lyme disease, travel, occupational, and leisure activity-related *Borrelia* exposures are quite likely in the study group, but were not documented as part of the intake process. Overall, our positivity rate is higher than other Canadian studies, but within range of American values for at-risk populations, which was anticipated considering the deliberate recruitment of individuals with Lyme and Lyme-like diseases. The under-detection of Lyme disease in Canada has also been described [65]. Discrepancies in the outcomes of different studies are influenced by the sampling design, geography of recruitment, and laboratory conducting the analysis. Significant inter-lab variability has been documented, and is a known drawback of two-tiered serology [66].

Notably, the agreement between test tiers appears to be lower in Canadian samples processed at Canadian reference labs than equivalent values originating from, and processed in, the United States. In the Nova Scotia serosurvey described above, none of the 215 individuals initially identified by whole-cell EIA subsequently returned a positive IgG WB through the National Microbiology Laboratory (NML) in Winnipeg, MN [62]. The NML is a central diagnostic processing facility that receives first-tier positive serum identified by provincial laboratories, in order to perform confirmatory two-tier testing. A retrospective analysis of reports generated by the NML identified that across all Canadian specimens reviewed under the purview of the study, 22% of positive C6 EIAs subsequently met CDC criteria for the IgG WB [67]. This value varied geographically, with the highest agreement between test tiers occurring in samples from Nova Scotia (52.1%), while New Brunswick registered only 9.1% [67]. The authors suggest that the low concordance could arise from false-positive C6 EIAs in regions with low environmental risk. Meanwhile, American studies have reported test-tier agreement above 60%, in some cases reaching 78% [68, 69]. Our observation that 25% of ELISA-positive specimens subsequently generated a positive WB is similar to the Canadian national average reported by the NML, but markedly lower than the agreement observed in American studies. Although our findings could be a consequence of lowering the pre-test probability by using

two-tiered serology indiscriminately as a surveillance tool, the samples under review at the NML were ordered for diagnostic purposes, indicating reasonable clinical suspicion of LD. This raises concerns that the diagnostic tests may not be optimized to detect the diversity of serotypes present in Canada. Accordingly, in a follow-up study of Canadian tick isolates, NML researchers identified strains that diverged considerably from the well characterized American specimens upon which the diagnostic tools were developed [70]. Further investigation is required to understand the implications of this divergence on the performance of serodiagnostics, and whether it contributes to the poor test-tier agreement or the low overall seroprevalence reported in high-risk zones. Ultimately, considerations of biodiversity should be central in the development of new laboratory diagnostic tools.

In the current study, seroreactivity was detected across all disease intake categories, and the IgG responsive cohort registered more profound symptom burden than IgM positive or seronegative individuals on select health scales, albeit with small sample sizes. Roughly half of the IgG seropositive findings occurred outside the presumptive LD intake category, however, the two groups were indistinguishable in the nature and burden of symptoms captured by our questionnaires. A major differentiating feature in the seropositive, presumptive LD group was the recollection of a tick bite / rash, and acute manifestations of the disease, which likely influenced the LD diagnosis. This emphasizes the continued clinical reliance on early objective signs of the disease, as well as the ambiguity of later presentations.

Subdividing the IgG seropositive cohort by reactive epitopes, we discovered that 47% (8/17) had antibodies against the 31 kDa OspA on the WB, in addition to the 5+ bands required to meet CDC criteria for test positivity. OspA was omitted from the WB scoring algorithm developed in the 1990s as it was considered a delayed epitope of little utility in the initial phases of illness [37], and was used as the immunogen in the first generation LD vaccine [71]. Indeed, development of an anti-OspA IgG response appears to be indicative of late disease in North America, appearing in *Borrelia* infections of greater than 6 months duration [72]. In contrast, OspA seroconversion is rare among European LD cases irrespective of the disease stage, paralleling geographic differences in genospecies associated with infection-related inflammation [73]. In the northeastern United States, OspA immunoreactivity has been detected in upwards of 65% of patients with Lyme arthritis (LA), an objective late manifestation of the disease [73]. It is particularly pronounced in treatment-refractory LA, wherein it has been implicated in autoimmune pathology among genetically predisposed individuals [43]. Anti-OspA immunoglobulins were also more frequent among patients who experienced generalized sequalae after antimicrobial intervention, including musculoskeletal pain and cognitive impairment, compared to those whose illness resolved with therapy [74]. On its own, however, anti-OspA IgG was not a reliable biomarker of PTLDS as it was not uniformly present in cases and absent in controls [74]. Notably, illness severity in the post-treatment group was not measured or stratified based on OspA response.

To that end, we believe that ours is the first study to consider an association between disease burden and OspA-IgG response within a seropositive, chronically ill cohort. While the group collectively reported profound, multisystem deficits that severely impacted daily functioning and quality of life, there was a clear distinction between OspA responders and non-responders. Those who lacked the 31 kDa band were among the most ill participants across all domains of assessment, whereas those with detectable OspA antibodies faired markedly better, although still considerably below a healthy baseline. Although we did not profile HLA haplotypes and cannot rule out arthritogenic processes, it appears that the disease presentation captured in the current investigation is phenotypically and mechanistically distinct from treatment-refractory Lyme arthritis, which is often a discreet and limited phenomenon [75] as opposed to the widespread systemic dysfunction reported by our cohort. It is tempting to speculate from our

observations that in the context of syndromic late Lyme, anti-OspA immunoglobulins bolster humoral defences to mitigate progressive infection. This aligns with substantial evidence that an anti-OspA response can confer protective immunity [76–78]. It would appear that the *Borrelia* genospecies predominantly dictates OspA epitope availability and immunoreactivity in late disease, while host predisposition determines whether an anti-OspA immune response is pathological or protective. Further investigation is warranted to assess the prognostic potential of this marker in a larger cohort.

Another striking finding to emerge from our study is the overlap between *Borrelia* seroreactivity and fibromyalgia (FMS), which has long been observed, but rarely formalized. FMS is characterized by widespread pain often accompanied by fatigue, unrefreshing sleep, and some degree of cognitive impairment [79]. The mechanistically neutral definition renders this a process of elimination diagnosis. Data from the 2010 Canadian Community Health Survey suggest that between 1.4% and 1.7% of the population has been diagnosed with FMS [80]. Although the hypothesis that LD triggers FMS and similar musculoskeletal disorders is not new, it largely failed to gain traction in the research sphere and remains underexplored. An association between the two diseases was first documented in the early days of LD characterization, when FMS was found to arise as consequence of progressive, untreated infection [19, 81]. These observations were swiftly followed by concerns that LD was being over diagnosed in a population that lacked established objective findings on the physical exam, or had already received standard treatment for LD (Hsu et al., 1993) [82]. Seroreactive results among patients with subjective symptoms were largely dismissed as false-positive findings, and antibiotic intervention was not recommended as the risk-benefit calculation was not favourable [83]. The CDC case definition at the time supported this perspective, although it has since evolved, and original treatment recommendations have been challenged by the observation that some late "syndromic" diseases proximal to *Borrelia* exposure appear to respond to antimicrobial intervention, albeit with a high relapse rate [20]. However, the significance of *Borrelia* in FMS specifically remains unclear in the literature. One small European study PCR-amplified a *Borrelia* chromosomal locus from muscle biopsies of half of their participants who suffered from chronic myalgias that persisted after treatment for classical presentations of LD [84]. More recently, however, *Borrelia* T cell reactivity was found not to differ between FMS and control subjects [85]. These discrepant findings justify larger systematic studies examining relationships between musculoskeletal disorders and vector-borne infection.

The broader interpretation and implications of LD serology also remain a matter of intense debate. Positive findings do not equate to active disease, and background seroprevalence is observed in endemic regions, as discussed above. Conversely, negative findings do not rule out LD, particularly in those treated for the disease or receiving immunomodulatory intervention. One study found that ~40% of patients who were diagnosed and treated for early objective LD failed to seroconvert according to IgM or IgG testing [86]. The authors therefore cautioned against requiring seropositivity as a diagnostic criterion for post-treatment Lyme. Conversely, concerns persist that positive laboratory findings in the context of atypical or subjective disease manifestations are unreliable [87]. IgG seropositivity in conjunction with non-specific symptoms like myalgia, brain fog, and fatigue that do not satisfy objective criteria for late Lyme may be classified as probable late Lyme disease [20] or chronic Lyme disease [21], yet the utility of testing individuals who lack objective clinical findings has been challenged [88]. Indeed, current clinical practice guidelines issued by the Infectious Diseases Society of America (IDSA) recommend against testing for *Borrelia* seroreactivity in patients with MS, amylotrophic lateral sclerosis (ALS), Parkinson's disease, dementia, or psychiatric and behavioural disorders [75]. Opposition to the broader application of serology beyond strict objective clinical criteria stems from the potential for cross reactive antibodies, and the increasing risk of false discovery when

the predictive value is lower. Yet, inherent to the concept of predictive value is a robust case definition or gold standard evidence of disease, which have arguably been lacking for LD. An artificially restrictive case definition that fails to account for the legitimate spectrum of manifestations will bias the interpretation of a test result. Likewise, limitations with laboratory diagnostics have stymied progress, and there is an unmet need for direct detection tools that can accurately report infection across all disease stages. Currently, there is no FDA approved test or biomarker to identify individuals with active infection who require antimicrobial therapy, and no way to monitor or ensure the eradication of pathogens. While academic medicine has largely dismissed chronic Lyme disease, the 1990s-era zeitgeist should be critically re-evaluated in the context of evolving technologies that improve biological and clinical resolution.

## Conclusions

Overall, our findings highlight the profound functional burden experienced by patients with presumptive Lyme and lookalike diseases. *Borrelia* IgG seropositivity in these groups should provide additional incentive to pursue mechanistic understanding of chronic and treatment-refractory illness with a goal of improving therapeutic options and patient outcomes.

## Supporting information

**S1 Fig. Principal component analysis matched to Fig 1B and 1C, except depicting individual participants by serological classification instead of intake category.**
(PDF)

**S1 File. Full data set for variables of interest.**
(XLSX)

**S1 Table. Mean rank values and Kruskal-Wallis H test statistics for the comparison of health outcomes by intake category.**
(PDF)

**S2 Table. Clinical characteristics of cohorts delineated by serology result.**
(PDF)

**S3 Table. Contingency tables exploring relationships between test results.**
(PDF)

**S4 Table. Concordance of individual immunoreactive IgG bands with the overall IgG serostatus (positive or negative by CDC criteria).**
(PDF)

## Acknowledgments

We extend our deep gratitude to the study participants who enabled this research. Victoria Rust and Emma Rogerson, undergraduate students at Mount Allison University, and Cindy McKinley-Brown, Research Nurse, Upper River Valley Hospital contributed to data collection; Laboratory Services, Upper River Valley Hospital were indispensable for sample collection and transfer. We are indebted to colleagues for providing feedback on drafts of this manuscript, and to data analysts at the University of Guelph McLaughlin Library for consulting on statistics. Errors and omissions are solely the responsibility of the authors.

## Author Contributions

**Conceptualization:** Victoria P. Sanderson, Jennifer C. Miller, Vladimir V. Bamm, Vett K. Lloyd, Gurpreet Singh-Ranger, Melanie K. B. Wills.

**Data curation:** Victoria P. Sanderson, Jennifer C. Miller, Vladimir V. Bamm, Manali Tilak.

**Formal analysis:** Victoria P. Sanderson, Vladimir V. Bamm, Manali Tilak, Melanie K. B. Wills.

**Funding acquisition:** Vett K. Lloyd, Gurpreet Singh-Ranger, Melanie K. B. Wills.

**Investigation:** Victoria P. Sanderson, Jennifer C. Miller, Vladimir V. Bamm, Gurpreet Singh-Ranger.

**Methodology:** Jennifer C. Miller.

**Project administration:** Victoria P. Sanderson, Gurpreet Singh-Ranger, Melanie K. B. Wills.

**Resources:** Gurpreet Singh-Ranger.

**Supervision:** Vett K. Lloyd, Gurpreet Singh-Ranger, Melanie K. B. Wills.

**Visualization:** Victoria P. Sanderson, Vladimir V. Bamm, Melanie K. B. Wills.

**Writing – original draft:** Melanie K. B. Wills.

**Writing – review & editing:** Victoria P. Sanderson, Jennifer C. Miller, Vladimir V. Bamm, Manali Tilak, Vett K. Lloyd, Gurpreet Singh-Ranger, Melanie K. B. Wills.

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
