## [Decision Letter · Decision Letter 0]

23 May 2023

PONE-D-23-06799Profiling disease burden and *Borrelia* seroprevalence in Canadians with complex and chronic illnessPLOS ONE

Dear Dr. Wills

Thank you for submitting your manuscript to PLOS ONE. After careful consideration, we feel that it has merit but does not fully meet PLOS ONE’s publication criteria as it currently stands. Therefore, we invite you to submit a revised version of the manuscript that addresses the points raised during the review process.

We look forward to receiving your revised manuscript.

Kind regards,

Raquel Inocencio da Luz, Phd

Academic Editor

PLOS ONE

Journal Requirements:

Reviewers' comments:

Reviewer's Responses to Questions

**Comments to the Author**

1. Is the manuscript technically sound, and do the data support the conclusions?

Reviewer #1: Yes

2. Has the statistical analysis been performed appropriately and rigorously? 

Reviewer #1: Yes

3. Have the authors made all data underlying the findings in their manuscript fully available?

Reviewer #1: No

4. Is the manuscript presented in an intelligible fashion and written in standard English?

Reviewer #1: Yes

5. Review Comments to the Author

Reviewer #1: M. Willis' study describes the association between several clinical conditions such as fibromyalgia, chronic fatigue syndrome and other complex syndromic pictures potentially associated with Lyme disease and the seroprevalence for Borrelia burgdorferi in the general population in eastern Canada.

The study has several strengths, including the use of multiple questionnaires to test for overlapping of different chronic conditions characterised by similar symptoms (chronic fatigue, arthralgias, muscle pain) and Lyme disease, and a recruitment technique aimed at recruiting different categories of subjects. The study design is well structured and the results obtained appear to be supported by the methodology used.

However, some revisions are necessary to improve the quality of the manuscript.

Methods:

The authors should more accurately specify the criteria for exclusion and inclusion of subjects recruited in the study. Furthermore, they should specify whether the STROBE checklist was used for cross-sectional observational studies.

In addition, they should specify whether the questionnaires used were validated for the Canadian population, and if so, the reference should be given.

Line 202-2011: Authors should describe in this section the statistical tests used, whether normality of the data was checked and whether parametric and non-parametric tests were performed. Furthermore, given the use of multiple questionnaires, static techniques such as a principal component analysis would be more appropriate to validate the results obtained

Discussion

In order to better understand the potential risk of exposure of the population under study, it would be appropriate to report information regarding the prevalence of Ixodes ricinus in the areas considered, and any data regarding seroprevalence in animals in the reference area, in the absence of seroprevalence data in the population.

A paragraph on study limitations should also be introduced. For example, several syndromes have been associated with other Borrelia species, such as Borrelia myamotoi, so assessing seroprevalence only for Borrelia burgorferi could represent a bias. Furthermore, some risk factors, such as occupational exposure to tick bites, do not seem to have been assessed by the authors, and this represents a further limitation of the study that should be reported.

6. PLOS authors have the option to publish the peer review history of their article (what does this mean?). If published, this will include your full peer review and any attached files.

Reviewer #1: No

---

## [Author Response · Author response to Decision Letter 0]

4 Aug 2023

Dear Editors and Reviewers,

Thank you for your thoughtful consideration of our manuscript, and invitation to resubmit a revised version. We have carefully considered the feedback, and have addressed each point as follows. Additions or modifications are annotated in the manuscript text with yellow highlighting. 

1 - The PLOS Data policy requires authors to make all data underlying the findings described in their manuscript fully available without restriction, with rare exception (please refer to the Data Availability Statement in the manuscript PDF file). The data should be provided as part of the manuscript or its supporting information, or deposited to a public repository.

The full dataset describing individual participants was not included in the initial submission as consent was not explicitly obtained from participants to publish personal findings. We have since revisited this point with research ethics boards that approved the protocol, and arrived at a compromise they deemed acceptable. The data that we now present in supplemental information (S1 Data File) include individual responses to survey questions, and results of laboratory findings. We have removed indirectly identifying information (participant ID, age, sex, specific diagnoses). 

2 - The authors should more accurately specify the criteria for exclusion and inclusion of subjects recruited in the study. Furthermore, they should specify whether the STROBE checklist was used for cross-sectional observational studies.

We have included a statement on line 126to acknowledge that STROBE guidelines were followed in the communication of these findings. 

Lines 133+ also clarify that the recruitment strategy was deliberately broad to capture a spectrum of complex and chronic illness, and individuals were only excluded on the basis of language (non-English speaker) or an inability to consent to research. 

3 - In addition, they should specify whether the questionnaires used were validated for the Canadian population, and if so, the reference should be given.

The three questionnaires have been extensively used in chronic disease research in North America. Although not specifically validated for Canadians, both the FIQ and SF-36 have been used and published in this population, and normative data was previously generated for the latter. We have updated the text to be more explicit about this (lines 154+). The HMQ was assessed on patients seeking treatment at clinics in the American Eastern Seaboard, and it is unclear whether this sample included Canadians. However, this region is an active destination for Lyme disease-related medical tourism. 

4 - Line 202-2011: Authors should describe in this section the statistical tests used, whether normality of the data was checked and whether parametric and non-parametric tests were performed.

We have updated the methods section (lines 213+) to include more information on the statistical tests run, and the basis for those decisions. We have also provided more context in the corresponding tables and figures (Table 1, Fig 1, S2 Table). Finally, since our multi-group hypothesis tests of continuous variables were performed using Kruskal-Wallis ANOVA on ranks (due to non-normally distributed response variables, and unequal distributions), we have added a supplemental table depicting the mean rank values (new S1 Table). 

5 - Furthermore, given the use of multiple questionnaires, static techniques such as a principal component analysis would be more appropriate to validate the results obtained.

To address this request, we have added an author to this manuscript who has training and experience in bioinformatics (Dr. Tilak) to oversee these additions. We used principal component analysis (PCA) to more thoroughly investigate health status according to intake category and serological test result. The methods are described in lines 225+; results are presented in lines 345, Fig 1B, Fig 1C, lines 409+ and S1 Fig. 

6 - In order to better understand the potential risk of exposure of the population under study, it would be appropriate to report information regarding the prevalence of Ixodes ricinus in the areas considered, and any data regarding seroprevalence in animals in the reference area, in the absence of seroprevalence data in the population.

We agree that this is pertinent information, and have further developed the discussion 

(lines 513+) to include surveillance data from the main vector tick, I. scapularis, as well as canine seroprevalence in the area. 

7 - A paragraph on study limitations should also be introduced. For example, several syndromes have been associated with other Borrelia species, such as Borrelia myamotoi, so assessing seroprevalence only for Borrelia burgorferi could represent a bias. Furthermore, some risk factors, such as occupational exposure to tick bites, do not seem to have been assessed by the authors, and this represents a further limitation of the study that should be reported.

We have expanded our discussion of testing limitations to emphasize that the study was not designed to detect Eurasian strains or the emerging pathogen B. miyamotoi, although all may be relevant to our target demographic (505+). We also note the risk posed by travel, occupation, and leisure activities in the region, and that such risk factors were not assessed as part of this study.

---

## [Decision Letter · Decision Letter 1]

29 Aug 2023

Profiling disease burden and *Borrelia* seroprevalence in Canadians with complex and chronic illness

PONE-D-23-06799R1

Dear Dr. 

We’re pleased to inform you that your manuscript has been judged scientifically suitable for publication and will be formally accepted for publication once it meets all outstanding technical requirements.

Kind regards,

Raquel Inocencio da Luz, Phd

Academic Editor

PLOS ONE

Additional Editor Comments (optional):

Reviewers' comments:

Reviewer's Responses to Questions

**Comments to the Author**

1. If the authors have adequately addressed your comments raised in a previous round of review and you feel that this manuscript is now acceptable for publication, you may indicate that here to bypass the “Comments to the Author” section, enter your conflict of interest statement in the “Confidential to Editor” section, and submit your "Accept" recommendation.

Reviewer #1: All comments have been addressed

2. Is the manuscript technically sound, and do the data support the conclusions?

Reviewer #1: Yes

3. Has the statistical analysis been performed appropriately and rigorously? 

Reviewer #1: Yes

4. Have the authors made all data underlying the findings in their manuscript fully available?

Reviewer #1: Yes

5. Is the manuscript presented in an intelligible fashion and written in standard English?

Reviewer #1: Yes

6. Review Comments to the Author

Reviewer #1: The authors adequately addressed the comments and the manuscript meets the criteria to be published in this journal.

7. PLOS authors have the option to publish the peer review history of their article (what does this mean?). If published, this will include your full peer review and any attached files.

Reviewer #1: No

---

## [Editor Report · Acceptance letter]

30 Oct 2023

PONE-D-23-06799R1 

Profiling disease burden and *Borrelia* seroprevalence in Canadians with complex and chronic illness 

Dear Dr. Wills:

I'm pleased to inform you that your manuscript has been deemed suitable for publication in PLOS ONE. Congratulations! Your manuscript is now with our production department. 

Kind regards, 

on behalf of

Dr Raquel Inocencio da Luz 

Academic Editor

PLOS ONE